# Experience of induction of labour: a cross-sectional postnatal survey of women at UK maternity units

Mairi Harkness ,[1] Cassandra Yuill,[2] Helen Cheyne ,[1] Christine McCourt,[2] Mairead Black,[3] Dharmintra Pasupathy,[4] Julia Sanders,[5] Neelam Heera,[6] Chlorice Wallace,[7] Sarah Jane Stock [8,9]

**Correspondence to**
Dr Mairi Harkness;
mairi.harkness@stir.ac.uk

## ABSTRACT

**Objectives** This study explored women's views and experiences of key elements of the induction of labour (IOL) process, including at home or in hospital cervical ripening (CR).

**Design** A questionnaire-based postnatal survey undertaken as part of the CHOICE Study process evaluation. The questionnaire was administered online and included fixed response and free-text options.

**Setting** National Health Service maternity units in the UK.

**Participants** 309 women who had an IOL.

**Outcome measures** The primary outcome measure was experience of IOL. Few women returned home during CR, meaning that statistical comparison between those who experienced home-based and hospital-based CR was not possible. Findings are reported as descriptive statistics with content analysis of women's comments providing context.

**Results** Information to support choice and understand what to expect about IOL is often inadequate or unavailable. Having IOL can create anxiety and remove options for birth that women had hoped would enhance their experience. Although it can provide a more comfortable environment, home CR is not always an acceptable solution. Women described maternity care negatively impacted by staffing shortages; delays to care sometimes led to unsafe situations. Women who had a positive experience of IOL described supportive interaction with staff as a significant contribution to that.

**Conclusions** Women do not experience IOL as a benign and consequence free intervention. There is urgent need for research to better target IOL and optimise safety and experience for women and their babies. Relatively few women were offered CR at home and further research is needed on this experience.

## BACKGROUND

Globally, induction of labour (IOL) rates have increased steadily over the last 20 years, with a recent surge in rates linked to improved evidence of safety and efficacy.[1] In the UK, around 30%–50% of births currently involve IOL, making it one of the most common obstetric interventions.[2 3]

Impact of IOL on women's experience of childbirth is unclear. Some evidence suggests

## STRENGTHS AND LIMITATIONS OF THIS STUDY

⇒ A robustly designed survey, including use of previously tested tools, was used to determine key aspects of women's experience of induction of labour.
⇒ Carefully considered recruitment strategies resulted in a large sample across multiple National Health Service sites.
⇒ Few women returned home during cervical ripening. As a result, data analysis produced descriptive, rather than inferential, statistics.
⇒ Qualitative analysis of women's free-text comments adds important context and aids understanding and interpretation of the findings.
⇒ The survey was conducted during the COVID-19 pandemic and findings should be considered within this unique context.

that IOl has little effect on overall satisfaction when compared with spontaneous labour,[1] however, undergoing IOL is understood to affect experience of childbirth: it is generally more painful than spontaneous labour; more likely to lead to additional interventions such as operative birth and epidural analgesia; and may remove the satisfaction of experiencing the more natural birth that many hope for.[4–6]

A positive birth experience is not merely nice to have. Women's experience during childbirth is described by the WHO as a 'critical aspect of ensuring high-quality labour and childbirth care.[7] Evidence underpins sociocultural and psychological aspects of care as significant for women during childbirth,[8] and negative experience of childbirth can be linked to serious psychological harm.[9] Despite this, there remains a dearth of evidence about women's experience of IOL.

The first stage of IOL, cervical ripening (CR), involves application of a drug or mechanical method to change a woman's cervix in preparation for labour. The second phase, if labour onset does not occur as a result of CR alone, is artificial rupture of the

fetal membranes (ARM) and intravenous administration of oxytocin. Traditionally, the whole process of IOL has been undertaken in hospital, however, some maternity units now offer home CR: women attend hospital for initial assessment and administration of CR agent and then return home for a period before labour starts or reassessment in hospital. Some evidence indicates that home CR could reduce duration of hospital stay during IOL and improve women's experience. There is increasing evidence to suggest that home CR is safe,[10–12] although its acceptability to women and impact on staff and maternity service has not been fully evaluated.[13]

The aim of this study was to explore women's views of their IOL, with a specific focus on their experiences of the initial stages of the process including CR at home and in-hospital.

## METHODS

### Design

This study was undertaken as part of the CHOICE Study, a prospective cohort study and process evaluation[8] commissioned by the National Institute for Health and Care Research to examine the safety, efficacy and acceptability of home CR and CR in hospital. The process evaluation (qCHOICE) included a postnatal questionnaire-based survey (reported here).

A cross-sectional online survey was used with the aim of describing women's views and experience of IOL, particularly those having CR at home and in hospital.

### Questionnaire development

The questionnaire was designed to explore key elements of the IOL process, using fixed response and free-text questions. The questionnaire assessed satisfaction, sense of control and mental well-being including previously tested tools.[14 15]

The IOL satisfaction questionnaire[14] was used to assess women's experiences of IOL, including information provision, anxiety, and physical and emotional discomfort. This questionnaire uses a five-point Likert scale (strongly agree–strongly disagree), we analysed this using N (%) agreement to create three categories: merging strongly agree with agree and strongly disagree with disagree.

A series of 10 questions from the Labour Agentry Scale (short)[15] were used to measure sense of control during childbirth. We used a six-point Likert type scale, and analysis was reported as percentage agreement across three categories: agree, neutral and disagree.

Demographic questions and questions about information and decision-making were based on those in the Scottish National Maternity Survey[16] altered to focus on IOL. The survey also included questions about cost of CR (home and hospital), including travel and childcare, for health economics evaluation to be reported elsewhere.

The questionnaire was pilot tested with the CHOICE Study personal and public involvement (PPI) group (women with recent experience of childbirth) and with maternal health researchers at City, University of London (11 people in total). Feedback was used to make minor changes to six questions, particularly those concerning decision-making and choice, improving clarity and accessibility. The scale used for the IOL satisfaction scale[14] was changed from 'never, rarely, some of the time, most of the time, always' to 'strongly agree, agree, unsure, disagree, strongly disagree'.

The survey was online, hosted by Online Surveys (www. onlinesurveys.ac.uk); a written version or completion via telephone and/or with a translator could be requested.

### Sample and recruitment

A convenience sampling approach was used, with women who underwent IOL at 37 weeks of pregnancy or more at the 21 National Health Service (NHS) sites participating in the CHOICE study potentially eligible. Those who experienced pregnancy loss were ineligible.

The planned sample size was calculated to enable comparison of the experiences of women who had home and hospital CR, as per the CHOICE Study aims.[13] The sample size required to compare the experiences of women who had home and hospital CR is estimated to be 89 per group (178 in total) for a probability of type 1 error set at 0.05 for a two-tailed comparison and a 80% power. This is based on use of the Labour Agentry Scale[15] where a change of 5.5 points is considered clinically meaningful.

The initial recruitment strategy was via electronic maternity notes system BadgerNet. We anticipated that women would receive information about the study using push notifications sent when an IOL was booked, and again at around 10 days postnatal when maternity care ended; directing them to view study information on their electronic maternity notes and access the study link. However, initial response rate was poor, it was not clear to what extent push notifications were being received, therefore, additional strategies were put in place while the survey remained open: first, efforts were targeted to seven sites (the five case study sites plus two that had expressed interest in the survey) to obtain a more focused response. At these sites a research midwife identified eligible participants on the postnatal ward and handed them a study card with a link to the online survey; In addition, a targeted social media advertising campaign (Facebook) was used in the five qCHOICE case study sites.

The questionnaire included initial eligibility screening questions, and ineligible respondents could not proceed to complete the survey.

The survey was open between February 2021 and April 2022. The planned sample calculation was subsequently revised when it became apparent that too few women were having CR at home for statistical comparison to be made. A sample size of 300 respondents was deemed practical, achievable and useful for the purpose of describing the experience of women who undergo CR.

## Patient and public involvement statement

The CHOICE Study PPI group were involved in development of participant information materials and survey recruitment strategies. The survey questionnaire was pilot tested with the group. A member of the PPI group is coauthor for this paper and will be involved in further dissemination of findings.

## Consent for participation

The questionnaire landing page included detailed participant information, researcher contact details for further information and consent questions to be completed before the survey could be accessed.

## Data analysis

Survey responses were exported from the online survey site into IBM SPSS Statistics V.23 software. Data were deidentified, cleaned and statistics produced.

We found that 12% (n=36) of the eligible survey respondents returned home during CR, therefore, it was not possible to use inferential statistics to compare their experience with those who remained in hospital. Instead, descriptive analysis was used across both groups with analysis of free-text responses providing context to the women's experience.

Free-text responses were analysed using thematic analysis approach; determining themes and, for some questions, how often those themes occurred. Initial analysis was undertaken by a single researcher, with emerging themes discussed and confirmed with three further members of the qCHOICE team, in an iterative process. NVivo V.12 software was used to organise the data and assist analysis.

## Findings

In total 320 questionnaires were completed. Nine responses were excluded as respondents had not had an IOL and a further two because their IOL happened prior to the CHOICE Study commencing. Three hundred and nine eligible responses were included in the analysis.

## Study respondents

Respondents had given birth in Scotland, England and Wales at 19 CHOICE Study sites and a further 6 NHS areas. Descriptive data for those who responded are given in table 1.

## Decision-making

The questionnaire included a series of questions about choice, decision-making and provision of information when offered IOL (table 2). Fifty seven per cent of women reported that they had either no choice, or no alternative option, about having IOL. While two-thirds (66%) felt that options were explained in a way they could understand, only half (50%) felt that they fully understood what to expect during IOL.

The free-text responses describe anxiety around risk to their baby's well-being as a major influence on the decision to accept the offer of an IOL. For some women, this risk was communicated in a way that that contributed to them feeling their choice about IOL was limited.

> I was induced because of my age. Whilst it was made clear that the decision was my choice, I also felt a lot of pressure from health professionals to be induced (Participant 010, Multip, Hospital CR)

> It was never something I had a choice in… I was told if I didn't get induced there was a high chance of my baby being stillborn because I was almost 42 weeks, so this scared me. (Participant 201, Primip, Home CR)

One hundred and forty-eight (48%) respondents stated that having an IOL changed their plans for labour and birth. Changes included: unable to use water immersion; change of planned place of birth to an obstetric unit from midwifery led unit (MLU) or home birth. Women also reported needing previously unwanted interventions including electronic fetal monitoring and intravenous oxytocin.

> I would have liked a water birth but was told it was no longer an option (Participant 080, Primip, Hospital CR)

### Time spent in hospital and at home during CR

Of the 266 respondents who had CR 39 (15%) were given the option to return home and 36 (14%) did return home. Of those, 22 (61%) had their IOL at a single maternity unit where home CR was offered to all women unless contraindicated.

Some women expressed disappointment at not having the option to return home, whereas others would not have wanted this option. Common themes were lack of choice about where CR occurred and feeling safer in hospital.

> I was told I could have balloon induction and go home at consultant appointment, then when I attended hospital was told this wasn't actually something I could have. (Participant 100, Primip, Hospital CR)

> I am pleased I didn't have a choice {of home CR} and stayed overnight. I did have a comfort I was in the right place. (Participant 081, Primip, Hospital CR)

For women who remained in hospital, the median duration in the antenatal area, between commencing CR and transfer to labour suite, was 22 hours. One hundred women (43%) reported being in antenatal area for 24 hours or longer and 42 (18%) for 48 hours or more. The longest duration of antenatal stay after CR commenced until transfer to a labour room was 260 hours; 11 days. Those who returned home remained at home for a median of 24 hours (range: 3–168 hours).

The respondents described delays at almost every stage of the IOL process. The most impactful was the wait to be transferred from antenatal area to labour ward after CR, either for ARM or because they were in labour. Staffing was frequently mentioned in relation to delays.

**Table 1** Description of the survey respondents

| First baby?<br>N=309 | Yes:<br>206 (67%) | No:<br>103 (33%) | | | | | |
|---|---|---|---|---|---|---|---|
| Maternal age<br>(years)<br>N=309 | Min: 19 | Max: 52 | Median: 31<br>SD: 4.993<br>Variance: 24.932 | | | | |
| Ethnicity<br>N=307<br>(2 missing) | White<br>291 (95%) | Asian/Asian<br>British<br>8 (3%) | Black:<br>4 (1%) | Mixed/multiple<br>ethnicity:<br>4 (1%) | | | |
| Social Deprivation<br>Index*<br>N=306 (3 missing) | 1<br>(most deprived)<br>61 (20%) | 2<br>57 (19%) | 3<br>60 (20%) | 4<br>73 (24%) | 4<br>(least deprived)<br>55 (18%) | | |
| Baby's birth<br>weight (grams)<br>N=297<br>(12 missing) | Min: 1790 | Max: 6600 | Median: 3500 | | | | |
| Gestation at IOL<br>(weeks)<br>N=309 | Min:37 | Max: 42 | Median: 39 | | | | |
| Reason for IOL<br>N=309 | Medical<br>(eg, raised blood<br>pressure)<br>146 (47%) | Post dates<br>70 (23%) | Size of baby<br>(large or small)<br>37 (12%) | Spontaneous<br>rupture of<br>membranes<br>20 (7%) | Reduced fetal<br>movements<br>19 (6%) | Other<br>14 (5%) | Don't know<br>3 (1%) |
| CR?<br>N=304<br>(5 missing) | Yes<br>266 (86%) | No<br>38 (12%) | | | | | |
| Method of IOL<br>N=309 | Prostaglandin gel/<br>pessary<br>202 (65%) | Balloon<br>catheter<br>43 (14%) | Non-CR methods:<br>membrane<br>sweep,<br>amniotomy,<br>intravenous<br>oxytocic<br>38 (12%) | Prostaglandin gel/<br>pessary and balloon<br>catheter<br>12 (4%) | Osmotic dilator<br>(eg, Dilapan-S)<br>9 (3%) | Don't know<br>5 (2%) | |
| Home CR<br>N=266 | Offered option to<br>return home<br>39 (15%) | Returned home<br>36 (13%) | | | | | |

*Social Deprivation Index quintiles: Scottish Index of Multiple Deprivation and English Indices of Deprivation (based on self-reported postcode).
CR, cervical ripening; IOL, induction of labour.

The staff were pushed to the brink which is why I was in hospital for 11 days before my waters were broken. (Participant 120, Primip, Hospital CR)

For some women the delay between the decision being made for IOL and the process being started, and subsequent delays after IOL commenced, conflicted with the information that their baby was at risk of death if the pregnancy continued and needed to be born soon.

When you have been told for 3 months that your baby could be in danger if you reach 39 weeks and then have to go beyond that because they don't have a bed for you, it's a very scary time (Participant 080, Primip, Hospital CR)

Some women described care being planned around service capacity rather than in line with guidance. At times this was described as having a direct effect on their IOL progress.

Had balloon induction 8am Monday. Balloon out 8am Tuesday and was 2–3 cm. However was sent home as there were not enough midwives to induce me further… Was taken back in on Thursday 4pm… 7am Friday taken to the delivery room… at that point was then back to 1cm. (Participant 194, Primip, Home CR)

Women often described poor experience of time spent in antenatal areas during CR: lack of privacy, lack of sleep, lack of food. They also reported a shortfall in support that midwives were able to provide before transfer to labour suite, manifested in lack of appropriate pain relief, lack of emotional support and concerns about clinical care.

I was labouring behind a curtain, no privacy, others all around me… It was really hard to focus and stay calm and relax with no privacy of my own, no pain relief and no food. (Participant 036, Multip, Hospital CR)

I spent 3 days crying in pain unable to eat or sleep in hospital (Participant 135, Primip, Hospital CR)

**Table 2** Choice, decision-making and information

| | |
|---|---|
| **Did you feel you were offered a choice about having your Labour induced or waiting for Labour to start?** | |
| Yes, I felt it was fully my decision | 122 (39%) |
| Yes, but I felt there was no other option | 117 (38%) |
| Not really, as I didn't have enough information | 10 (3%) |
| No, I didn't feel I was given a choice | 60 (19%) |
| **Were these options explained to you in a way that you could understand?** | |
| Yes, I felt I fully understood | 205 (66%) |
| Partly | 70 (23%) |
| Not really | 19 (6%) |
| I'm not sure | 2 (0.6%) |
| No | 13 (4%) |
| **Did you get enough information about what to expect during induction of labour?** | |
| Yes, I felt I fully understood | 155 (50%) |
| Partly | 90 (29%) |
| Not really | 38 (12%) |
| I'm not sure | 0 |
| No | 26 (8%) |
| **Did having an induction lead to any change in your birthplace plans?** | |
| Yes | 148 (48%) |
| No | 153 (59%) |
| I'm not sure | 8 (3%) |

Among women who stayed in hospital throughout the induction process 196 (74%) had a birth partner who stayed with them compared with 40 (98%) women who returned home. Free-text responses indicated that when CR happened in hospital birth partners were not always able to stay as often as women wanted.

> It was a lonely experience, my husband was not allowed to come in until I was in active labour (Participant 137, Multip, Hospital CR)

Presence of their birth partner was very important to the respondents, with their absence described as absence of important support. It was also reported that exclusion of birth partners, usually the other parent, denied them full participation in an important life experience.

> the induction also meant my husband actually missed our son being born because I progressed so quickly (Participant 225, Multip, Hospital CR)

> My partner and I feel like one of the most important experiences of our lives was stolen from us (Participant 264, Primip, Hospital CR)

Twenty-eight women described being in established labour for a prolonged period and/or approaching second stage while remaining in the antenatal area.

> I was told I couldn't have [ epidural analgesia l] until I moved to labour ward but I couldn't move to labour ward as it was full. I was only moved when I was pushing (Participant 113, Primip, Hospital CR)

### Experience of labour induction

The findings suggest that for many women IOL, regardless of any time spent at home, is a period of anxiety, pain and discomfort, and of feeling powerless and lacking control (table 3). Over one-third of women (101) (38% who remained in hospital and 44% of those who went home) did not feel comfortable with their decision to have an IOL, while 22% (21% who remained in hospital and 31% who returned home) were worried that IOL might not be safe. Although 36% of those who went home reported anxiety about this, in relation to future choice of home or hospital for CR, more than half in each case (55% hospital and 64% home) said they would choose the same option again.

While 67% of women who stayed in hospital reported having good family support throughout the induction, this was 97% for those who went home.

Findings related to aspects of participants sense of control are described (table 4). Overall around one-third of women reported feeling like a failure, 40% felt powerless; about half felt fearful and around half felt confident or in control. Most women (both home and hospital) reported feeling that they were with people who cared about them.

Respondents' feelings of anxiety, powerlessness and lack of control were apparent in free-text comments:

> I felt like things happened to me rather than being part of any decisions (Participant 109, Primip, Hospital CR)

> I felt that choices were taken away from me… I don't think I was given enough information… I wasn't told whether this [painful CR] was normal (Participant 113, Primip, Hospital CR)

Forty-one women described their experiences of IOL as traumatic and/or having caused significant long-term negative impact on their physical and/or mental well-being.

> It was all so horrendous that I will never have another child. It gives me anxiety thinking about it all. Before this experience I did want more than one child (Participant 184, Primip, Hospital CR)

Thirty-five women described experiences that were positive overall. Supportive interaction with staff made a significant contribution to women's positive experiences, as did feeling 'safe' and 'cared for'.

> Supportive staff, well informed and felt every decision was genuinely done for our wellbeing (Participant 092, Primip, Hospital CR)

**Table 3** Satisfaction during CR and IOL (Henry et al[14])

| From initiation of CR to admission to labour suite | Women who remained in hospital N=227 (3 missing) | | Women who returned home N=36 (0 missing) | |
|---|---|---|---|---|
| | Agree | Unsure and disagree | Agree | Unsure and disagree |
| I felt a lot of discomfort | 143 (63%) | 84 (37%) | 28 (78%) | 8 (22%) |
| I was able to cope with the discomfort | 155 (68%) | 72 (32%) | 29 (81%) | 7 (19%) |
| I felt anxious about being in hospital /going home | 115 (51%) | 112 (49%) | 14 (39%) | 22 (61%) |
| I was able to relax on the AN ward/at home | 101 (44%) | 126 (56%) | 20 (56%) | 16 (44%) |
| I was able to rest on the AN ward/home | 103 (45%) | 124 (55%) | 24 (67%) | 12 (33%) |
| I had good family support in hospital/home | 151 (67%) | 76 (33%) | 35 (97%) | 1 (3%) |
| I had easy access to information from the staff | 127 (56%) | 100 (44%) | 23 (64%) | 13 (36%) |
| I was worried the induction might not be safe | 47 (21%) | 180 (79%) | 11 (31%) | 25 (69%) |
| I would have preferred to go home/stay at the hospital | 97 (43%) | 130 (57%) | 12 (33%) | 24 (67%) |
| I felt embarrassed by the catheter or gel | 21 (9%) | 206 (91%) | 3 (8%) | 33 (92%) |
| While at home I felt anxious about being at home not hospital | N/A | N/A | 13 (36%) | 23 (64%) |
| IOL | N=230 | | N=36 | |
| I felt anxious about being induced | 168 (73%) | 62 (26%) | 31 (86%) | 5 (14%) |
| I felt in control | 62 (27%) | 168 (73%) | 9 (25%) | 27 (75%) |
| I understood what was happening | 174 (76%) | 56 (24%) | 24 (67%) | 12 (33%) |
| I felt relaxed | 62 (27%) | 168 (72%) | 8 (22%) | 28 (78%) |
| Everything made sense | 137 (60%) | 93 (40%) | 18 (50%) | 18 (50%) |
| I was given clear information | 151 (66%) | 79 (34%) | 17 (47%) | 19 (53%) |
| I felt comfortable with my choice about my care | 145 (63%) | 85 (37%) | 20 (56%) | 16 (44%) |
| I had access to information about the types of induction available | 119 (52%) | 111 (48%) | 20 (56%) | 16 (45%) |
| I had easy access to information about what to do | 122 (53%) | 108 (47%) | 19 (53%) | 17 (48%) |
| I found the induction process uncomfortable | 144 (63%) | 86 (38%) | 32 (89%) | 4 (11%) |
| I was worried about when my labour would begin | 176 (76%) | 54 (22%) | 26 (72%) | 10 (28%) |
| I would choose staying in hospital /going home again | 126 (55%) | 104 (45%) | 23 (64%) | 13 (36%) |
| I would recommend staying in hospital during induction/going home to other women | 125 (54%) | 105 (36%) | 22 (61%) | 14 (39%) |

CR, cervical ripening; IOL, induction of labour; N/A, not available.

**Table 4** Sense of control during induction of labour

| | Agree All respondents N=309 | Agree In-hospital CR N=230 | Agree Home CR N=36 |
|---|---|---|---|
| I felt tense | 188 (61%) | 143 (62%) | 23 (64%) |
| I felt important | 215 (69%) | 149 (65%) | 28 (78%) |
| I felt confident | 149 (48%) | 102 (44%) | 19 (53%) |
| I felt in control | 140 (45%) | 98 (43%) | 18 (50%) |
| I felt fearful | 152 (49%) | 113 (49%) | 19 (53%) |
| I felt relaxed | 95 (31%) | 68 (30%) | 10 (28%) |
| I felt good about my behaviour | 243 (79%) | 183 (80%) | 27 (75%) |
| I felt helpless (powerless) | 122 (39%) | 92 (40%) | 15 (42%) |
| I felt like a failure | 96 (31%) | 71 (31%) | 12 (33%) |
| I felt I was with people who care about me | 249 (81%) | 179 (78%) | 31 (86%) |

CR, cervical ripening.

"My midwife {name} was incredibly supportive throughout labour and birth. We felt safe and cared for." (Participant 040, Primip, Hospital CR)

## DISCUSSION

This study reports on the experience of women undergoing CR and IOL more generally, describing wide variation. While some women report a positive experience, significant numbers described a negative experience and a small but important number had an experience that was so traumatic they wished to avoid future, previously planned, pregnancies.

The IOL process begins when a pregnant woman and her caregiver first discuss IOL; facilitation of informed decision-making is integral to quality maternity care and prominent within current National Institute for Health and Care Excellence (NICE) guidance around IOL.[4] However, decision-making about IOL may be complex and for many of the women in this survey it seemed poorly supported: most respondents (60%) felt that they either had no choice about IOL or no alternative option. Communicating risk in relation to IOL can be difficult and contentious,[17] and this study found that communication around IOL led some women to believe that induction was required to avoid an otherwise high chance of their baby dying. Informed decision making must be underpinned by good quality information, and clinicians should include absolute as well as relative risks of stillbirth when sharing information with women.

Provision of antenatal education and information are recognised as key factors in shaping women's expectations and their ability to cope with labour and birth.[18] However, just half (50%) of the survey respondents felt that they fully understood what to expect during their IOL and almost one-third (32%) felt unable to cope with the discomfort of CR. Active decision-making may contribute to positive experience when women require previously unwanted interventions,[19] however, the women surveyed here described an absence of real choice about IOL alongside significant restrictions on options for care when they accept induction. The most reported restrictions on birth plans were accessing an MLU and use of water during labour or birth. Both are known to improve experience: water is an effective method of pain relief during the first stage of labour;[4] births planned in MLUs are associated with significantly reduced intrapartum interventions, with no difference in neonatal outcome[20 21] and increased satisfaction with care.[22 23]

Free-text responses reveal further impact on experience of care, with women describing a paradox of deciding, or sometimes being persuaded, to have an IOL because of perceived risk but then facing an absence of urgency to commence induction and significant delays during the process itself. This was a marked aspect of negative experience and caused stress and anxiety, especially when women reported having been told they required induction because their baby was at risk of stillbirth.

Few women were given the option to return home after CR commenced, limiting our ability to explore whether home CR may improve experience. Of the small number (just 36 women) who did more than half would recommend this option, most were able to cope with the discomfort they felt and most felt able to rest and relax during CR. This is despite most also having had CR using mechanical methods which may be associated with increased initial pain.[6 14] Nonetheless, one-third of women who went home would have preferred to stay in hospital and 36% felt anxious about being at home rather than in hospital.

Those who experienced CR in hospital describe an environment that frequently failed to meet their needs. Time spent in hospital, during CR and prior to transfer to labour ward, was often characterised by inadequate support from staff and absence of birth partners, combined with insufficient pain relief, lack of privacy in shared wards and failure to take seriously or listen to women's concerns about pain, discomfort and labour progress.

Delays were reported at almost every stage of the IOL process, the most impactful being late assessment of progress and application of further CR agent, and long delays when ready for transfer to labour ward. The women associated delays with poor staffing, reflecting similar recent experience of UK maternity services.[24 25] It was not unusual for respondents to feel that their physical safety was compromised; some reporting eventual transfers to delivery suite during advanced labour including second stage. Thus, women who had been informed their baby was at increased risk of death were receiving care below that required once in established labour.

Physical safety and psychological well-being are equally valued by women[19] and some respondents reported that they felt the care they received compromised both. Mental ill health is a leading and increasing cause of maternal morbidity and mortality.[26] That around half the women (49%) in this survey reported feeling fearful during their IOL, and that many described an experience that was traumatic with lasting negative impact, is of significant concern.

It is of note that there is extensive literature on women's negative experience but much less on the nature of women's positive experience.[8] Some women who responded to the survey did have positive experiences of IOL, and the features they describe offer insight to aid understanding of how best to support women undergoing IOL. The most significant factor in women's positive experience was their interaction with staff. This echoes longstanding knowledge of the importance interaction with caregivers holds for women's experience,[18] and is something that individual practitioners can influence despite organisational and workforce factors.

The overwhelming majority of respondents (81%) stated that they felt that they were with someone who cared about them, but concerningly, nearly 20% of women did not feel this way. Compassionate and

respectful care encompasses a sense of care as genuine through 'emotional availability'[8] and it is encouraging that this appears to have been facilitated for most. For many women enabling birth partner attendance during the difficult and often lengthy period of CR would be a simple and effective means of further supporting this.

## Limitations

The sampling and recruitment strategies employed meant that it was not possible to determine a denominator from which a response rate could be calculated. It is recognised that this may introduce bias among the characteristics and experiences of those who chose to respond. However, adoption of a pragmatic approach to achieve a large sample across multiple NHS sites was deemed to outweigh potential limitations. The results are descriptive and not intended to be generalisable.

The COVID-19 pandemic presented significant challenges, both to study recruitment and to the context in which the respondents received the maternity care they were describing. Maternity unit policies and practices changed in response to various stages of the pandemic, with at least one of the five case study sites halting the offer of home IOL completely. Some placed severe restrictions on the presence of partners on antenatal wards, although it is of note that prepandemic it was often usual practice to restrict birth partner attendance to visiting hours only. The survey findings should be understood within this context.

The work was undertaken in the UK at a time when NHS maternity services were under significant strain, and experiencing a significant rise in induction rates. While there is no doubt that this context impacts the findings and their interpretation, it was clear that factors unrelated to staff shortages were also influencing the experiences reported here. In addition, findings such as lack of informed choice, pain and anxiety, fear and concerns about lack of monitoring or support until in active labour have been identified in studies conducted prior to current staff shortages.[6] Similar difficulties also impact provision of maternity services in many countries worldwide and, without significant systemic and economic change, is likely to remain the context in which IOL is offered for the immediate future.

A significant limitation was that so few women were offered the option to return home during CR (n=36, 12%), limiting the opportunity to compare their experience with those who remained in hospital. However, there is very little information about women's experience of home CR and this work adds important and relevant understanding about women's experience of undergoing IOL, CR in particular, both in hospital and at home.

## CONCLUSION

This work shows that women undergoing IOL do not experience it as a benign, consequence free intervention; IOL often causes anxiety and removes options for birth

that women had hoped would enhance their experience and outcomes. Assessing risks and benefits when offering, and considering the offer of, an IOL can be complex; clinicians must ensure that women are informed of and understand the absolute as well as relative risks of continuing the pregnancy, along with the risks and consequences of IOL itself.

Experience of childbirth is important to women; known to influence physical, social and psychological short-term and long-term outcomes. Although some women had a positive experience of IOL, many experienced poor care, inadequate communication and delays during IOL that had the potential to jeopardise their safety or that of their baby. For some, this led to long-term psychological maternal morbidity including the desire to avoid future pregnancies.

Returning home during CR may be an option to improve women's experience of IOL; however, few women were offered this opportunity and numbers were too small to draw definite conclusions.

Women's experience of childbirth was profoundly affected by staffing and resource issues; this context makes it difficult to extrapolate poor experience of IOL from poor experience of childbirth due to lack of staff and subsequent inadequate care.

IOL rates have been increased with good intentions of reducing rates of stillbirth and severe neonatal morbidity, but this does not always appear to have been accompanied by adequate planning or increase in the facilities and resources that are necessary to provide an effective service. In addition, the findings of this study indicate that greater attention to the quality of information giving to underpin informed choice is needed. This accords with a recent report on risk assessment in maternity pathways in the UK, which calls for more individualised risk assessments.[27]

It is crucial that the expected principles of person-centred individualised care, provided with dignity and respect, apply equally to women experiencing IOL. Listening to women and service users, ensuring that practice is based on their needs, and that services are sufficiently resourced are all essential to the provision of safe and effective induction pathways.

**Author affiliations**
[1]Nursing Midwifery and Allied Health Research Unit, University of Stirling, Stirling, UK
[2]Centre for Maternal and Child Health Research, City University of London, London, UK
[3]Aberdeen Centre for Women's Health Research, Aberdeen Maternity Hospital, Aberdeen, UK
[4]Faculty of Medicine and Health, The University of Sydney, Sydney, New South Wales, Australia
[5]School of Healthcare Sciences, Cardiff University, Cardiff, UK
[6]Cysters, Birmingham, UK
[7]Princess Royal University Hospital, Orpington, UK
[8]The University of Edinburgh Usher Institute of Population Health Sciences and Informatics, Edinburgh, UK
[9]The University of Edinburgh MRC Centre for Reproductive Health, Edinburgh, UK

**Contributors** MH: substantial contribution to the design of the work, and acquisition, analysis, and interpretation of data for the work, drafted the work and revised it critically for important intellectual content, final approval of the version to be published, agreed to be accountable for all aspects of the work in ensuring that questions related to the accuracy or integrity of any part of the work are appropriately investigated and resolved. Responsible for overall content as guarantor. CY: substantial contribution to the design of the work, and acquisition, analysis, and interpretation of data for the work, revised the work critically for important intellectual content, final approval of the version to be published, agreed to be accountable for all aspects of the work in ensuring that questions related to the accuracy or integrity of any part of the work are appropriately investigated and resolved. HC: substantial contribution to the design of the work, and acquisition, analysis, and interpretation of data for the work, revised the work critically for important intellectual content, final approval of the version to be published, agreed to be accountable for all aspects of the work in ensuring that questions related to the accuracy or integrity of any part of the work are appropriately investigated and resolved. CM: substantial contribution to the design of the work, and acquisition, analysis, and interpretation of data for the work, revised the work critically for important intellectual content, final approval of the version to be published, agreed to be accountable for all aspects of the work in ensuring that questions related to the accuracy or integrity of any part of the work are appropriately investigated and resolved. MB: substantial contribution to the conception and design of the work, and interpretation of data for the work, revised the work critically for important intellectual content, final approval of the version to be published, agreed to be accountable for all aspects of the work in ensuring that questions related to the accuracy or integrity of any part of the work are appropriately investigated and resolved. DP: substantial contribution to the conception and design of the work, revised the work critically for important intellectual content, final approval of the version to be published, agreed to be accountable for all aspects of the work in ensuring that questions related to the accuracy or integrity of any part of the work are appropriately investigated and resolved. JS: substantial contribution to the conception and design of the work, and interpretation of the data for the work, revised the work critically for important intellectual content, final approval of the version to be published, agreed to be accountable for all aspects of the work in ensuring that questions related to the accuracy or integrity of any part of the work are appropriately investigated and resolved. CW: substantial contribution to the design of the work, revised the work critically for important intellectual content, final approval of the version to be published, agreed to be accountable for all aspects of the work in ensuring that questions related to the accuracy or integrity of any part of the work are appropriately investigated and resolved. NH: substantial contribution to the design of the work, revised the work critically for important intellectual content, final approval of the version to be published, agreed to be accountable for all aspects of the work in ensuring that questions related to the accuracy or integrity of any part of the work are appropriately investigated and resolved. SJS: substantial contribution to the conception and design of the work, and interpretation of data for the work, revised the work critically for important intellectual content, final approval of the version to be published, agreed to be accountable for all aspects of the work in ensuring that questions related to the accuracy or integrity of any part of the work are appropriately investigated and resolved.

**Funding** The CHOICE Study was funded by the National Institute of Healthcare Research Health Technology and Assessment (NIHR HTA) NIHR127569.

**Disclaimer** The views expressed are those of the authors and not necessarily those of the NIHR or the Department of Health and Social Care.

**Competing interests** The authors declare the following competing interests Support for the present manuscript: The CHOICE Study was funded by National Institute of Healthcare Research Health Technology and Assessment (NIHR HTA). The grant was paid to institution and the authors have the following associations: Patient and public involvement (PPI) Group member (paid); Funded research fellows (2); Principal investigator and coinvestigators with funded time. CHOICE study budget has been used by some authors for travel and conference cost. Authors hold additional, paid to institution, grants from: NIHR; Wellcome Trust; Medical Research Council; Chief Scientist Office of Scotland; Tommy's Charity; Scottish government; Aberlour Childcare Trust (small project grant); University of Stirling article processing fund. Authors declare the following: Consultancy fees: Natera, consultancy on preterm birth treatments (paid to institution); Honoria: Hologic, Honoria for educational talk (paid to institution); Expert testimony: Expert witness in (midwifery) in civil litigation claims (self employed) Authors declare the following participation on data safety monitoring or advisory boards: Membership of several NIHR Trial Steering committees; NIHR Health Technology and Assessment DMC and TSC Authors are unpaid members or trustees of the following Boards and Committees: National Institute of Health and Care Excellence Antenatal Guideline Group 2018-202; Cysters (CIO) Chair of trustees; Platform Housing Group, trainee Board role; UK Government Period Poverty Task Force, member; UK sepsis Trust, Trustee.

**Patient and public involvement** Patients and/or the public were involved in the design, or conduct, or reporting, or dissemination plans of this research. Refer to the Methods section for further details.

**Patient consent for publication** Not applicable.

**Ethics approval** This study involves human participants and was approved by York & Humber—Sheffield Research Ethics Committee. June 2020 (IRAS: 276788). Participants gave informed consent to participate in the study before taking part.

**Provenance and peer review** Not commissioned; externally peer reviewed.

**Data availability statement** Data are available on reasonable request.

**ORCID iDs**
Mairi Harkness http://orcid.org/0000-0002-1007-7648
Helen Cheyne http://orcid.org/0000-0001-5738-8390
Sarah Jane Stock http://orcid.org/0000-0003-4308-856X

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
