## [Reviewer comments · BMJ Open]

This paper was submitted to a another journal from BMJ but declined for publication following peer review. The authors addressed the reviewers' comments and submitted the revised paper to BMJ Open. The paper was subsequently accepted for publication at BMJ Open.

ARTICLE DETAILS

TITLE (PROVISIONAL)	Experience of induction of labour: a cross-sectional postnatal survey of women at UK maternity units
AUTHORS	Harkness, Mairi; Yuill, Cassandra; Cheyne, Helen; McCourt, Christine; Black, Mairead; Pasupathy, Dharmindra; Sanders, Julia; Heera, Neelam; Wallace, Chlorice; Stock, Sarah

VERSION 1 – REVIEW

REVIEWER	Gonzalez-de la Torre, Hector Universidad de Las Palmas de Gran Canaria, Enfermeria
REVIEW RETURNED	28-Jan-2023

GENERAL COMMENTS	Dear Authors Thank you for allowing me to review this manuscript. I congratulate you for the work you have done and for initiating the CHOICE research project. In order to improve the manuscript, I would like to make some suggestions and ask you about some issues that may not have been sufficiently clear. I also make some notes on formal editing aspects that need to be confirmed by the editor. -In the section What is already known on this topic and What this study adds, perhaps it would be better not to use the acronym IOL or CR, which I think would improve readers' understanding. -I consider that the choice of keywords could be improved. I make my proposal:(Mesh terms: satisfaction, patient/ Labor, Induced/ Cervical Ripening/ Pregnant women/ Maternal Health Services) -The introduction seems to me to be correct, although perhaps it could have gone deeper by providing more data and references on induction rates, not only in the UK, but also in the rest of the world, as this phenomenon is global (also in my country). Furthermore, there are studies carried out on birth satisfaction that have found no statistically significant differences between spontaneous births and inductions of labour with respect to satisfaction. In my opinion you should at least state that there is controversy about this (as you admit in your discussion). This aspect is important to reflect in the introduction, as it is directly related to the aim of this study. It would also have been interesting to add some additional reference to cervical ripening at home, as the reference you cite is from your study protocol. Although it is true that very little has been published (hence the interest of this manuscript), there are some published studies (here are three examples).
--

(Stock SJ, Taylor R, Mairs R, Azaghdani A, Hor K, Smith I, Dundas K, Kissack C, Norman JE, Denison F. Home cervical ripening with dinoprostone gel in nulliparous women with singleton pregnancies. *Obstet Gynecol.* 2014 Aug;124(2 Pt 1):354-360. doi: 10.1097/AOG.0000000000000394.)-

Some of you are the authors of this article

(Agarwal K, Batra A, Batra A, Dabral A, Aggarwal A. Evaluation of isosorbide mononitrate for cervical ripening prior to induction of labor for postdated pregnancy in an outpatient setting. *Int J Gynaecol Obstet.* 2012 Sep;118(3):205-9. doi: 10.1016/j.ijgo.2012.04.017.)

(Reid M, Lorimer K, Norman JE, Bollapragada SS, Norrie J. The home as an appropriate setting for women undertaking cervical ripening before the induction of labour. *Midwifery.* 2011 Feb;27(1):30-5. doi: 10.1016/j.midw.2009.11.003. Epub 2010 Jan 4. PMID: 20045584.)

Method

This study is part of a macro-study (CHOICE). Please excuse me if some of the issues I allude to are already explained in the published protocol of this study.(Stock SJ, Bhide A, Richardson H, Black M, Yuill C, Harkness M, Reid M, Wee F, Cheyne H, McCourt C, Rana D, Boyd KA, Sanders J, Heera N, Huddleston J, Denison F, Pasupathy D, Modi N, Smith G, Norrie J. Cervical ripening at home or in-hospital-prospective cohort study and process evaluation (CHOICE) study: a protocol. *BMJ Open.* 2021 May 4;11(5):e050452. doi: 10.1136/bmjopen-2021-050452.)

Questionnaire

The measurement instruments are not validated and this is perhaps one of the major limitations of this study, although in principle you accept and acknowledge this limitation. Perhaps you could provide some psychometric data if available from the two tools from which you have developed this one.

Using two Likert scales of different sizes in a questionnaire (5 and 6 responses/ the 3-response scale may have a mean response bias compared to the 6-response scale) is uncommon and can only be understood in the context of trying to use scales that have already been validated and not changing them. I would have opted to unify the scores, but this is my personal opinion conditioned by my line of research focused on questionnaire validation.

Please indicate at least the number of women you used for the pre-test. Please also indicate how many questions and which ones you modified.

Sample and recruitment

There is no sample calculation. In my opinion a sample calculation should have been done based on some aspect (% of inductions with CR, difference in satisfaction scores, etc.).

It might be interesting to provide more information on NHS sites, in order to help establish the external validity of the study for other researchers in other countries.

Online surveys are currently a very popular method of data collection, but the validity of the results obtained through them must be ensured. In this case, what control mechanisms were put in place to ensure that only women who met the inclusion criteria answered the survey?

Analysis and Findings

I think there is margin for improvement.

In the analysis of the quantitative, the variable age I do not understand why you did not calculate a measure of dispersion

	(since you use the median as a measure of central tendency you must use the interquartile range). In table 1 the variable Social deprivation index appears, can you explain what this variable consists of? -You should elaborate on the qualitative analysis of the free-response questions. Indicate how many people participated in the analysis and whether there were any data triangulation procedures. When you present the results it is not clear how many themes emerged; perhaps if the editor would consider a table or graph with the themes it would improve understanding in this respect. In this table the verbatims that support the themes could be shown. -In the case of the analysis of satisfaction during CR and IOL and sense of control during induction of labour, the items of the scales were done on the basis of % agreement and disagreement. In my opinion this can be done, but it would have been more appropriate to calculate the Likert scale scores, with their means and standard deviations. This would have allowed a difference of means and effect size to be calculated, regardless of the fact that there were only 36 participants in a single group. Explain the rationale for this form of analysis Dicussion Correct Conclusion I think that say that: "This work demonstrates that women undergoing IOL do not experience it as a benign, consequence free intervention;" is excessive.It must be change the verb demonstrate for other like "show" or other similar I congratulate the authors for this research and hope that my contributions will improve the quality of the manuscript.
--	--

REVIEWER	Gould, Alexander J. Brown University
REVIEW RETURNED	21-Feb-2023

GENERAL COMMENTS	Thank you for this work. As IOL--including elective IOL--becomes more common, qualitative research that assesses patient satisfaction is very valuable. My major comment is that many of the patient complaints highlighted in this work stem from a lack of resources at the sites providing IOL and are not intrinsic limitations of the IOL process itself. I think it would be helpful for clarity to state this more clearly. The authors did describe "staffing shortages" as a negative impact even in the abstract, and discuss a lack of resources throughout the conclusion. So, while the current paper provides a good assessment of IOL satisfaction within the current (flawed) system at the study sites, it is generalizable only to sites with similar staffing issues. To amend this shortcoming, this should be more clearly stated in limitations. Similarly, the authors may consider suggesting that future research be first aimed at quality improvement to address these known gaps, and then reassessment of IOL satisfaction once appropriate and safe care is attainable. They might also consider offering explanations for these staffing issues/lapses in standards of care, and what might be needed to solve them. If the ultimate goal of this research is to understand patient experience of IOL, and the outcome has shown that patient experience is limited profoundly by staffing and resource (e.g. bed availability)
---

	shortages, then better understanding the causes of the resource limitations will also help providers understand how to optimize patient experience of IOL. Separately, elective induction was not listed as an indication for induction, while some patients reported induction indications such as age. Were none of these elective inductions?
--	--

VERSION 1 – AUTHOR RESPONSE

Reviewer One	
Dr. Hector Gonzalez-de la Torre, Universidad de Las Palmas de Gran Canaria	
Introduction	
In the section What is already known on this topic and What this study adds, perhaps it would be better not to use the acronym IOL or CR, which I think would improve readers' understanding.	This section has been removed on editor request, as above
I consider that the choice of keywords could be improved. I make my proposal:(Mesh terms: satisfaction, patient/ Labor, Induced/ Cervical Ripening/ Pregnant women/ Maternal Health Services)	Thank you. We have included these key words.
The introduction seems to me to be correct, although perhaps it could have gone deeper by providing more data and references on induction rates, not only in the UK, but also in the rest of the world, as this phenomenon is global (also in my country).	Thank you. The word count is very limited, but we agree that this is an important point and have added text at the beginning of the introduction to address this.
Furthermore, there are studies carried out on birth satisfaction that have found no statistically significant differences between spontaneous births and inductions of labour with respect to satisfaction. In my opinion you should at least state that there is controversy about this (as you admit in your discussion). This aspect is important to reflect in the introduction, as it is directly related to the aim of this study.	Thank you. We believe one of the strengths of our study is description of overall experience of IOL, with satisfaction a component of that experience. We agree that there is some controversy in the evidence around this and have added text within paragraph two of the introduction.
It would also have been interesting to add some additional reference to cervical ripening at home, as the reference you cite is from your study protocol. Although it is true that very little has been published (hence the interest of this	Thank you, again. We have updated the text and referencing in paragraph four of the introduction to reflect this comment.

manuscript), there are some published studies (here are three examples).

(Stock SJ, Taylor R, Mairs R, Azaghdani A, Hor K, Smith I, Dundas K, Kissack C, Norman JE, Denison F. Home cervical ripening with dinoprostone gel in nulliparous women with singleton pregnancies. *Obstet Gynecol.* 2014 Aug;124(2 Pt 1):354-360. doi: 10.1097/AOG.0000000000000394.)-

Some of you are the authors of this article

(Agarwal K, Batra A, Batra A, Dabral A, Aggarwal A. Evaluation of isosorbide mononitrate for cervical ripening prior to induction of labor for postdated pregnancy in an outpatient setting. *Int J Gynaecol Obstet.* 2012 Sep;118(3):205-9. doi: 10.1016/j.ijgo.2012.04.017.)

(Reid M, Lorimer K, Norman JE, Bollapragada SS, Norrie J. The home as an appropriate setting for women undertaking cervical ripening before the induction of labour. *Midwifery.* 2011 Feb;27(1):30-5. doi: 10.1016/j.midw.2009.11.003. Epub 2010 Jan 4. PMID: 20045584.)

Method

This study is part of a macro-study (CHOICE). Please excuse me if some of the issues I allude to are already explained in the published protocol of this study.

Questionnaire

The measurement instruments are not validated and this is perhaps one of the major limitations of this study, although in principle you accept and acknowledge this limitation. Perhaps you could provide some psychometric data if available from the two tools from which you have developed this one Using two Likert scales of different sizes in a questionnaire (5 and 6 responses/ the 3-response scale may have a mean response bias compared to the 6-response scale) is uncommon and can only be understood in the context of trying to use scales that have already been validated and not changing them. I would have opted to unify the scores, but this is my personal opinion conditioned by my line of research focused on questionnaire validation.	Thank you, this is a very reasonable point. There is no validated questionnaire, that we are aware of, to assess satisfaction during IOL. The tool used by Henry et al (2013), although not formally validated, has been successfully pre-tested. The Labour agency scale has been validated for psychometric properties. We decided on these tools as they were those most closely aligned with our population and aims. Although perhaps not ideal, we feel that the tools used within the questionnaire were appropriate particularly given that the analysis reported here is descriptive.
Please indicate at least the number of women you used for the pre-test. Please also indicate how many questions and which ones you modified.	A total of 11 people pre-tested the survey questionnaire in two phases during 2020. Feedback led to minor modification of wording for questions related to planned place of birth, choice, method of cervical ripening, expenses and costs associated with home vs. hospital CR. There were additional minor changes to wording in the Henry questions. The scale used for the latter was changed from “Never, Rarely, Some of the time, Most of the time, Always” to “Strongly agree, Agree, Unsure, Disagree, Strongly disagree”. Six questions were changed overall. The main tools/questions within the survey had been used previously (Henry et al, the labour agency scale, and the Scottish Maternity Survey) and so had been pre-tested previously.

	Text has been added at: Questionnaire development, paragraph 5.
Sample and recruitment	
There is no sample calculation. In my opinion a sample calculation should have been done based on some aspect (% of inductions with CR, difference in satisfaction scores, etc.).	Thank you, the editor raised a similar query. Here is our response: Sample size was initially calculated on the basis that a comparison would be possible between home and hospital CR. It became apparent during the study that this would not be possible because of small number of people returning home. A full explanation of this, along with original sample size calculation has been added at Sample and Recruitment.
It might be interesting to provide more information on NHS sites, in order to help establish the external validity of the study for other researchers in other countries.	We have added brief context of the UK NHS maternity services in which the study was conducted.
Online surveys are currently a very popular method of data collection, but the validity of the results obtained through them must be ensured. In this case, what control mechanisms were put in place to ensure that only women who met the inclusion criteria answered the survey?	Thank you, text has been added under 'sample and recruitment', paragraph four, to explain the process for ensuring eligibility of participants.
Analysis and Findings	
In the analysis of the quantitative, the variable age I do not understand why you did not calculate a measure of dispersion (since you use the median as a measure of central tendency you must use the interquartile range)	Standard deviation and variance have been added to the figures given for maternal age in Table one.

In table 1 the variable Social deprivation index appears, can you explain what this variable consists of?	A footnote has been added to Table one explaining the social deprivation scores.
You should elaborate on the qualitative analysis of the free-response questions. Indicate how many people participated in the analysis and whether there were any data triangulation procedures. When you present the results it is not clear how many themes emerged; perhaps if the editor would consider a table or graph with the themes it would improve understanding in this respect. In this table the verbatims that support the themes could be shown.	Text has been added under sub-heading 'data analysis', paragraph three.
In the case of the analysis of satisfaction during CR and IOL and sense of control during induction of labour, the items of the scales were done on the basis of % agreement and disagreement. In my opinion this can be done, but it would have been more appropriate to calculate the Likert scale scores, with their means and standard deviations. This would have allowed a difference of means and effect size to be calculated, regardless of the fact that there were only 36 participants in a single group. Explain the rationale for this form of analysis	The form taken to report the scores was based on the reporting style used by Henry et al (2013), the only comparable work that describes satisfaction with IOL.
Conclusion	
I think that say that: "This work demonstrates that women undergoing IOL do not experience it as a benign, consequence free intervention;" is excessive. It must be change the verb demonstrate for other like "show" or other similar	Thank you, we concur and have changed this wording.
Reviewer 2 Dr. Alexander J. Gould, Brown University	
My major comment is that many of the patient complaints highlighted in this work stem from a lack of resources at the sites providing IOL and are not intrinsic limitations of the IOL process itself. I think it would be helpful for clarity to state this more clearly. The authors did describe "staffing shortages" as a negative impact even in the abstract, and discuss a lack of resources throughout the conclusion. So, while the current paper provides a good assessment of IOL satisfaction within the current (flawed) system at the study sites, it is generalizable only to sites with similar staffing issues. To amend this shortcoming, this should be more clearly stated in limitations.	Thank you, this is an important point. A paragraph has been added within the Limitations section that we hope addresses your comment.

Similarly, the authors may consider suggesting that future research be first aimed at quality improvement to address these known gaps, and then reassessment of IOL satisfaction once appropriate and safe care is attainable. They might also consider offering explanations for these staffing issues/lapses in standards of care, and what might be needed to solve them. If the ultimate goal of this research is to understand patient experience of IOL, and the outcome has shown that patient experience is limited profoundly by staffing and resource (e.g. bed availability) shortages, then better understanding the causes of the resource limitations will also help providers understand how to optimize patient experience of IOL.	Thank you. In addition to the paragraph added to the Limitations section, we have added text within the conclusion that we hope further addresses this important point.
Separately, elective induction was not listed as an indication for induction, while some patients reported induction indications such as age. Were none of these elective inductions?	This is an interesting point. The survey gave the following options in answer to this question:  • Length of pregnancy • Medical reasons • I'm not sure • Other All of those who responded 'other' gave reasons other than maternal choice for IOL, for example: maternal age, SROM, big baby or small baby. No respondents indicated that maternal choice alone was the reason for IOL.

VERSION 2 – REVIEW

REVIEWER	Gonzalez-de la Torre, Hector Universidad de Las Palmas de Gran Canaria, Enfermeria
REVIEW RETURNED	02-Apr-2023

GENERAL COMMENTS	Dear authors Thank you for the changes made. You have done a good job in modifying the manuscript. I recommend that in further work on the topic you improve the data collection instrument, so that the analysis will be more complete. For my part, your study can be published. Best regards.
---

REVIEWER	Gould, Alexander J. Brown University
REVIEW RETURNED	06-Apr-2023

GENERAL COMMENTS

Thank you very much for your attention to these reviewer comments. I agree that the manuscript is improved after revisions. Congratulations on your work!